

# Effective field theory for quasicrystals and phasons dynamics

**Matteo Baggioli[1*] and Michael Landry[2]**

**1** Instituto de Fisica Teorica UAM/CSIC, c/Nicolas Cabrera 13-15,
Universidad Autonoma de Madrid, Cantoblanco, 28049 Madrid, Spain
**2** Department of Physics, Center for Theoretical Physics,
Columbia University, 538W 120th Street, New York, NY, 10027, USA

★ matteo.baggioli@uam.es

## Abstract

We build an effective field theory (EFT) for quasicrystals – aperiodic incommensurate lattice structures – at finite temperature, entirely based on symmetry arguments and a well-define action principle. By means of Schwinger-Keldysh techniques, we derive the full dissipative dynamics of the system and we recover the experimentally observed diffusion-to-propagation crossover of the *phason mode*. From a symmetry point of view, the diffusive nature of the phason at long wavelengths is due to the fact that the internal translations, or phason shifts, are symmetries of the system with no associated Noether currents. The latter feature is compatible with the EFT description only because of the presence of dissipation (finite temperature) and the lack of periodic order. Finally, we comment on the similarities with certain homogeneous holographic models and we formally derive the universal relation between the pinning frequency of the phonons and the damping and diffusion constant of the phason.



# 1  (Long) Introduction

> "How can you govern a country which has 246 varieties of cheese ?"

*Charles de Gaulle*

All matter is made of atoms yet, the diversity of the phases in which matter can be realized is both beautiful and astonishing. The most basic phases of matter, solid, liquid and gas have been realized and discussed since ancient history. A naive classification argument simply relies on putting them into a container and observing whether they take its shape and volume. A solid holds its shape and volume, a liquid takes the shape of its container but retains a fixed volume, and a gas expands to any size, taking both the shape and the volume of the container.

For a more complete understanding of the possible phases of matter, this pragmatic criterion is not satisfactory and a deeper, more fundamental theoretical distinction has to be found. Moreover, there are many more phases of matter which are not taken into account in this trilogy, which evade the standard criteria of classification. Two such examples are glasses [1, 2] and topological materials [3, 4]. Additionally, the distinction between solids and liquids (and phases of matter in general) is not so neat and it depends crucially on the length-scale and time-scale of consideration[1].

Classifying phases of matter—whether they are exotic or ordinary—is a full-time job for every theoretical physicists and it continues to yield surprising results. An elegant means of classifying states of matter relates to the concept of universality and renormalization group flow introduced by Wilson [7]. In this view, a large variety of different materials, with completely different microscopic features can be described by theories that "flow" down to the same low-energy descriptions. These low-energy descriptions are solely determined by a small subset of (relevant) operators and by a specific set of preserved and spontaneoulsy broken symmetries. From this Wilsonian perspective, symmetries are the key-elements to understand the different standard[2] phases of matter. A simple example is the Landau classification of second order phase transitions [8] as the separation between two phases with different symmetry-breaking patterns, e.g. metal/superfluid, ferromagnet/antiferromagnet, etc.

The proper and modern formal language to describe and understand the different phases of matter is that of effective field theory (EFT) [9, 10]. The general idea is that phases of condensed matter always spontaneously break the Poincaré group simply because their equilibrium states selects a preferred reference frame, i.e. the frame in which the sample of matter is

---

[1]A concrete example is the experimental observation of propagating shear waves at low frequencies (solid-like behaviour) in liquids [5], whose theoretical nature is surprising and under investigation [6].

[2]By "standard" we mean phases of matter which can be distinguished using symmetries. In this manuscript, we will restrict ourselves to this subclass and we will ignore more complex phases such as topological ones and more complicated phase transitions (e.g. quantum phase transitions, metal-insulator transitions, etc) which cannot be described within the Landau framework.

stationary. In this picture, the classification of the different phases is in 1-to-1 correspondence with the classification of the possible symmetry-breaking patterns of the Poincaré group. At zero temperature, the standard field theory methods are available and the formal construction has been presented using the coset techniques in [11] and later generalized in [12]. In this low-energy description the fundamental dynamical degrees of freedom are the Goldstone bosons corresponding to the specific symmetry breaking pattern (e.g. the phonons in a solids [13]). Heavier massive modes are integrated out and they do not appear in the EFT description[3].

These methods are very powerful and have been employed in a plethora of systems [18–23]; nevertheless the freedom from microscopics comes at a price. The first, and obvious, drawback is the impossibility of computing any transport coefficients (or in the EFT language, Wilson coefficients). As a concrete example, EFT methods can yield the dispersion relations of the phonons in a solids but they will never give you any information about their speeds. A second, and perhaps more severe difficulty, arises when trying to extend the EFT methods to describe finite-temperature systems. In particular, standard action principles and field theory formulations do not allow for dissipation, which is an important feature of all finite-temperature systems, e.g. fluids. After some preliminary perturbative solutions [24], two tools acquired a predominant role in this direction: holography [25] and Schwinger-Keldysh (SK) techniques [26]. In this work, we will take inspiration from some recent results in holography and we will apply SK techniques to achieve an EFT description of quasicrystals at finite temperature.

Although Goldstone's theorem is very well understood in Poincaré-invariant systems that exhibit spontaneous symmetry breaking (SSB) of *internal* symmetries (see [27] for a review), when the underlying theory is not Poincaré-invariant or when Poincaré symmetry is spontaneously broken, the phenomenology becomes richer and the physics more complicated. In these situations, the Goldstone modes may exhibit more unusual dispersion relations (i.e. it is no longer necessary that $\omega \sim k$), and the number of Goldstones may be less than the number of spontaneously broken generators. Using non-relativistic EFT techniques, situations in which dispersion relations are not linear (i.e. $\omega \sim k^2$) can be understood in terms of so-called type II Goldstone modes [28, 29]. And when spacetime symmetries are broken, extraneous Goldstones can be removed with so-called Inverse Higgs constraint, thus allowing for fewer Goldstones than broken generators [29, 30].

Recently, the existence of diffusive Goldstone bosons has been advocated in the context of open-dissipative systems [31–33]. These theoretical expectations have been confirmed in simple holographic systems with spontaneously broken translations [34–41]. Diffusive Goldstones, known as *phasons*, are also know to be present in aperiodic crystals—quasicrystals. See the next section for a brief discussion and several references.

The construction of a finite temperature EFT for quasicrystals and the understanding of the diffusive phasons and their dynamics is the main focus of this paper. We will combine the techniques of [12] and [42] to build a formal theoretical construction for quasicrystals which is complementary to their hydrodynamic description [43] and it relies totally on the symmetries of the system. Although the literature on quasicrystals abounds with statements such as "*Mode counting arguments and the Goldstone theorem lead to the prediction that phason modes are diffusive-like excitation,*" we have not been able to find a single satisfactory formal explanation based on symmetries. With the present paper, we aim to remedy that situation.

---

[3]A counterexample to this statement is the case of soft explicit breaking, which gives rise to pseudo-Goldstone modes with a parametrically small mass gap (e.g. pions) [14–17].

## 1.1 What is a Quasicrystal?

Quasicrystals, or quasiperiodic structures, are materials with perfect long-range order but lacking periodicity [44–47]. From an experimental point of view, the structure of materials is investigated using diffraction experiments (electrons, X-rays or neutrons) which display intensity distributions directly related to the (Fourier transform) of the material structure. In this sense, long-range order is equivalent to the presence of a finite number of Fourier components with sharp diffraction peaks. From this point of view (see right panel of Fig.1), quasicrystals are quite similar to standard periodic crystalline structures and they are fundamentally opposite to amorphous materials such as glasses, which lack any form of long-range order ($\equiv$ no sharp Bragg peaks). Nevertheless, quasicrystals differ from ordinary crystals because of the absence of periodicity. Ordinary crystalline lattices are unchanged under a discrete translation

$$\vec{x} \rightarrow \vec{x} + \vec{V}_i \,, \tag{1}$$

where $\vec{V}_i$ for $i = 1, 2, 3$ are the lattice vectors defining the unit cell of the periodic structure. By definition, periodicity means that all the points which are separated by such vectors are totally identical. From a more theoretical perspective, periodic crystals break translational invariance down to a discrete subgroup, while quasicrystals break the full continuous translational symmetry.

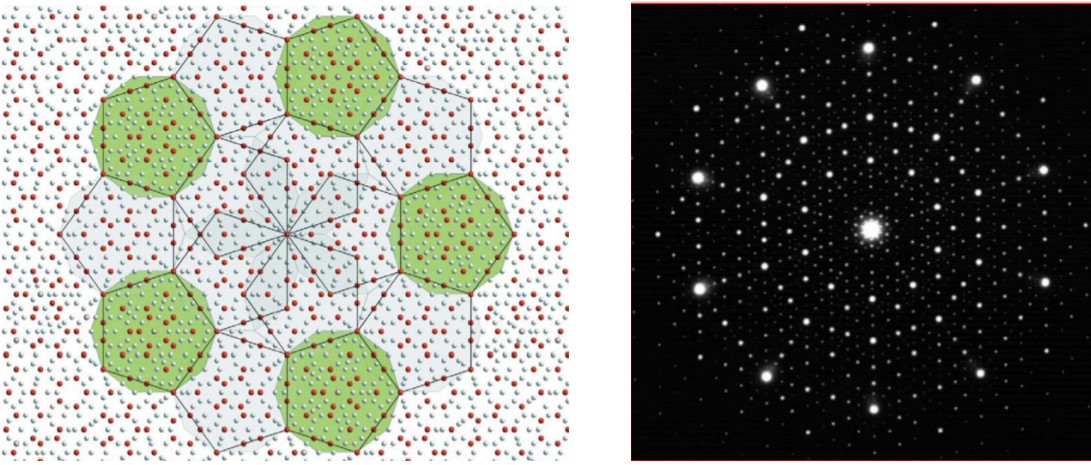

Figure 1: **Left:** A section perpendicular to the decagonal axis of the Al-Co-Ni quasicrystal from [48]. **Right:** Electron diffraction pattern from an icosahedral quasicrystal. Notice that the intensity distribution of the diffraction pattern varies over many orders of magnitude.

As shown in Fig.1, aperiodic quasicrystalline structure are usually associated with the presence of discrete rotational symmetries which are incompatible with periodicity.[4] Interestingly, some of them display even a discrete form of scale (and conformal) invariance [49].

A simple argument to convince ourselves that long-range order does not necessitate periodicity is the following. Consider a 1D array of atoms organized in a periodic distribution with lattice size $a$. The corresponding density is given by:

$$\rho(x) = \sum_n \delta(x - n a) \,, \tag{2}$$

---

[4]Five-fold rotations and rotations of order over six-fold are forbidden by periodic space tiling.

where $n$ is an integer. The Fourier transform displays a periodic structure as well, given by

$$\mathcal{F}_h = \sum_h \delta\left(k_h - 2\pi\frac{h}{a}\right),\tag{3}$$

and labelled by a single integer index $h$. Clearly, this example will exhibit both long-range order and periodicity. Let us now superimpose a second periodic structure, whose lattice size is incommensurate with the previous one. More precisely, let us consider the following distribution:

$$\rho(a) = \sum_n \delta(x - na) + \sum_m \delta(x - \alpha ma).\tag{4}$$

It is evident that, if $\alpha$ is not a rational number, then the resulting structure is not periodic. Its Fourier transform cannot be defined using a single index $h$. Nevertheless, it possesses long-range order and it displays sharp Bragg peaks. This is the simplest example of an incommensurate structure, which is indeed a specific case of aperiodic crystalline structures.

The mass density of a periodic lattice can always be expressed as:

$$\rho(\vec{x}) = \frac{1}{V}\sum_{\vec{G}} \rho(\vec{G})\,e^{i\,\vec{G}\cdot\vec{r}},\tag{5}$$

with $G$ the vector of the reciprocal lattice which can always be decomposed into a complete basis as:

$$\vec{G} = n_1\hat{x}_1 + n_2\hat{x}_2 + n_3\hat{x}_3,\tag{6}$$

where $\hat{x}_n$ span the full reciprocal space. In these notations, the position of a lattice point is simply described by the triad $\{n_1, n_2, n_3\}$.

In a quasicrystal, this description is insufficient. The vector $\vec{G}$ describing the position of the lattice points in reciprocal space cannot be decomposed as in Eq.(6) but rather as:

$$\vec{G} = \underbrace{n_1\hat{x}_1 + n_2\hat{x}_2 + n_3\hat{x}_3}_{d\ \text{physical dimensions}} + \underbrace{\sum_{i=d}^{D-d} n_i\hat{x}_i}_{D-d}.\tag{7}$$

In other words, more fundamental vectors than physical dimensions are needed to determine the position of the lattice points in an aperiodic structure. In the case of an icosahedral phase, we need for example 6 of them.

This point is fundamental in the description of quasicrystals and it suggests immediately that aperiodic structures can always be seen as periodic structures in an extra-dimensional space. The number of extra-dimensions needed depends on the specifics of the quasicrystal symmetries. Quasicrystals can always be obtained using projection/cut operators from higher-dimensional periodic structures.

This description is known as the *superspace description* [50]. One example for a $1D$ quasicrystal is displayed in Fig.2. Let us consider a two-dimensional square lattice with lattice spacing $= a$, defined by the following distribution:

$$\rho(x,y) = \sum_{n,m} \delta(x - a)\,\delta(x - ma).\tag{8}$$

This corresponds to a two-dimensional reciprocal lattice with wave-vector $2\pi/a$. Let us now perform a one-dimensional cut on this lattice defined by a single angle $\alpha$ with respect to the

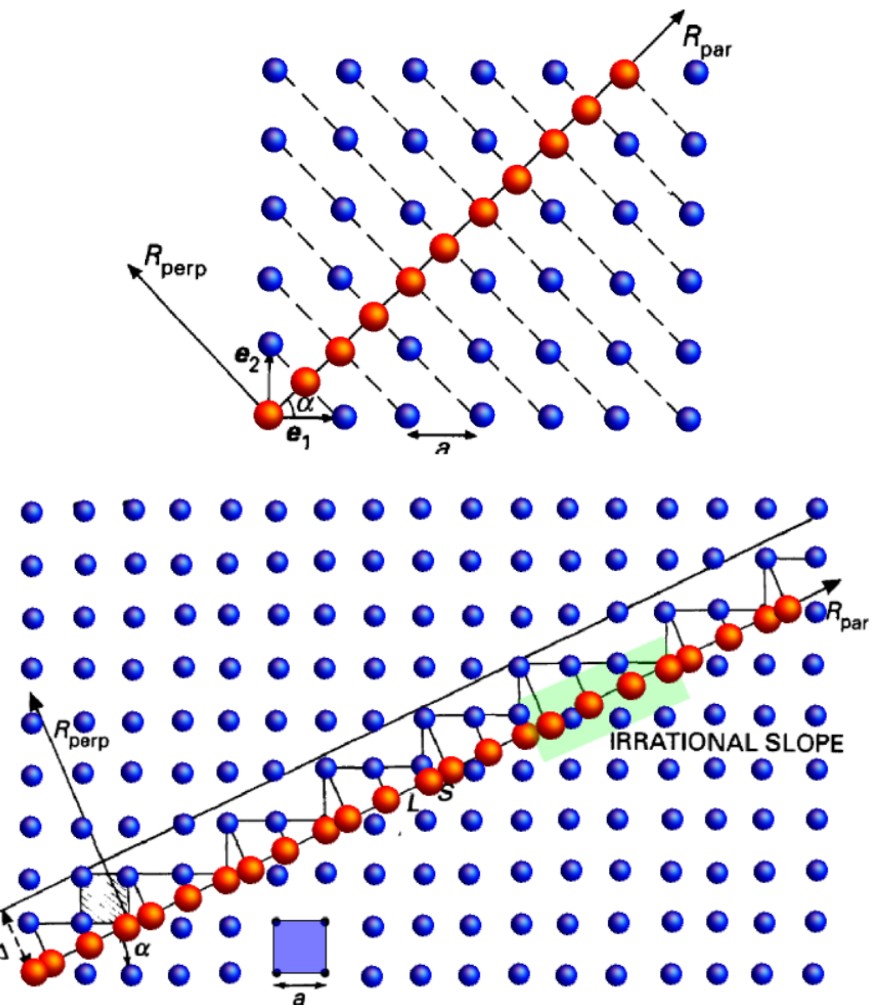

Figure 2: The *superspace* description for quasicrystals in a $2D \rightarrow 1D$ example. The blue dots are the positions of the atoms in the extra-dimensional periodic crystal; the red ones are the position of the atoms in the quasicrystal. The periodic 2D lattice has lattice spacing $a$. The figures are adapted from [47]. **Top:** A rational cut producing a periodic 1D crystal. **Bottom:** An irrational cut producing an aperiodic-crystal made of two lattice sizes $L, S$ incommensurate between them.

horizontal axes of the lattice. If the angle is rational (e.g. $45°$ as in the top panel of Fig.2), then the resulting 1D structure, obtained by projecting the two-dimensional lattice points on the line, will be periodic as well, with period $l = a \cos \alpha$. Now, let us to choose an irrational angle $\alpha$. In that case, the 1D resulting structure[5] (see bottom panel of Fig.2) is not periodic anymore and it is composed by two types of tiles (long and short) whose lengths are given by:

$$L = a \cos \alpha, \qquad S = a \sin \alpha. \tag{9}$$

In the example of Fig.2 we find an aperiodic sequence:

$$\dots SLLSLLLSLLS \dots \tag{10}$$

If we take, $\cos \alpha / \sin \alpha = \tau$, the golden mean, it is easy to see that the LS sequence obeys a Fibonacci sequence. This superspace description will play a fundamental role in the analysis

---

[5]The technical prescription consists in projecting the 2D lattice points in a strip of thickness $\Delta = a(\cos \alpha + \sin \alpha)$ into the 1D cut.

of the low-energy excitations in a quasicrystal.

Before moving to the next topic, we will briefly review an interesting case of aperiodic structure: the modulated crystal. Take a two-dimensional lattice structures whose atomic positions are defined by the vectors

$$r_n = a \, (n_1 \, \hat{x}_1 + n_2 \, \hat{x}_2) \,, \tag{11}$$

and apply a modulation to it such that the new positions become

$$r_n = a \, ((n_1 + \epsilon \sin(q \, n_1 \, \alpha)) \, \hat{x}_1 + n_2 \, \hat{x}_2) \,, \tag{12}$$

where $q = 2\pi/a$ is the modulation vector. As before, for angles $\alpha$ which are irrational, the structure becomes aperiodic and takes the name of incommensurate modulate structure. A $d-$dimensional modulate structure can again be seen as the intersection of a $d + 1-$ dimensional periodic structure with the $d-$dimensional physical space. Importantly, the phase of the modulated function can be chosen arbitrarily. Changing the phase corresponds to rearranging the atoms in a way that the equilibrium free energy remains unmodified. A continuous shift of the modulation creates an infinite set of indistinguishable configurations which can be visualized by piling them up on an axis perpendicular to the physical directions. This perpendicular direction is called *phase space* and it is exactly analogous to the perpendicular direction in the *superspace* description for quasicrystals explained above and depicted in Fig.2. For the sake of our discussion, modulated structures and quasicrystals share the same physics[6].

The dynamics in the perpendicular direction gives rise to the so-called *phason mode,* which will be the protagonist of our next section.

In the interest of brevity, we have discussed only the quasicrystal features which will be necessary for our discussion. We refer the more interested readers to [51–55].

## 1.2 Phasons dynamics

In the previous section, we outlined the principal differences between a periodic crystalline structure and an aperiodic one. In particular, we have seen that the lack of periodicity can be interpreted as the existence of an extra internal dimension—the transverse direction in the *superspace* formalism or the phase space in the modulated structures. Given this extra ingredient, which is absent in periodic structures, it is normal to expect an additional dynamical mode to appear in quasicrystals beyond the usual phonons of standard crystals. Such a mode is associated with displacement in the extra internal dimension and it is commonly referred to as the *phason*.

Complete discussions regarding quasicrystal dynamics and the properties of the phason mode can be found in [56–62]. Here, we will recall only the key features. We start by spoiling the end of the story. Quasicrystals, differ from periodic structures in that they display an additional Goldstone mode called the *phason*. The name was coined in 1971 by Overhauser [65] in the context of charge density wave (CDW) systems. In the regime of long wave-lengths (or equivalently small momenta), the phason mode displays a diffusive dispersion relation:

$$\omega = -i \, D \, k^2 + \dots \,, \tag{13}$$

---

[6]Strictly speaking, this is true only in one dimension. In higher dimensions there are subtle differences between the two. See [47].

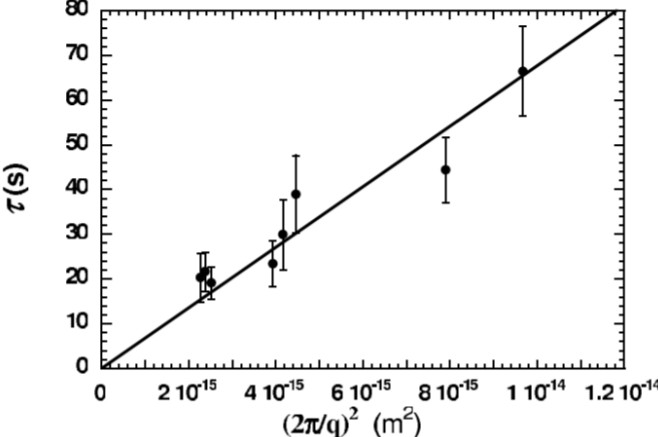 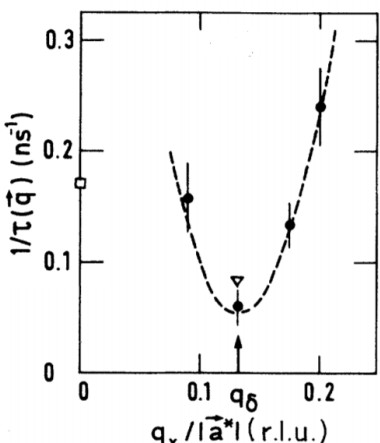

Figure 3: Experimental evidences for the diffusive nature of the phasons. **Left:** Phason relaxation time $\tau$ in function of the momentum $q$ in the i-AlPdMn icosahedral quasicrystal at $T = 650°$ C from [63]. **Right:** Relaxation rate $\tau^{-1}(q)$ for $NaNO_2$ at $T = T_{incomm.} + 0.55K$ from [64].

which is experimentally observed [63, 64, 66] (see Fig.3). Phason modes may be responsible for several distinctive finite-temperature properties such as the deviation of the heat capacity from the Dulong-Petit law observed experimentally in a quasicrystalline Al-Pd-Mn alloy [67]. Moreover, (I) the diffusion constant $D$ grows with temperature and it vanishes at zero temperature; (II) at large momenta, the dispersion relation turns into a linear propagating mode $\omega = v k + \cdots$, with $v$ set by a specific new elastic constant. The transition is estimated around 100 Angstroms. Thus, we would expect that light scattering experiments would probe mainly the diffusive regime, whereas inelastic neutron scattering experiments would probe mainly the propagating regime. We will come back to this point later.

The rest of this section will be devoted to understand these points and their physical origin in detail.

In order to understand the nature of the phason mode it is convenient to go back to the superspace formalism shown in Fig.2. As we have already explained, a quasicrystal can be described using a set of "parallel" coordinates, which correspond to the physical dimensions, and a set of (in our example just one) transverse coordinates, which represents the extra-dimension in the aforementioned internal space. Likewise, the dynamics can be split into these two orthogonal sets. The displacements in the parallel directions are the standard physical displacements, which gives rise to the phonons. The additional transverse displacements are known as phason displacements and are particular to quasicrystals. These displacements do not corresponds to infinitesimal shifts of the atomic positions. On the contrary, they correspond to *atomic flips* or *atomic jumps*. The idea (see Fig.4) is that the free energy is invariant under a rigid shift along the transverse directions and therefore a gapless hydrodynamic mode will be associated with this process. Nevertheless, a rigid shift in the transverse direction has the effect of flipping some atomic positions in the physical space. More concretely, an internal shift is associated with a flip of the kind:

$$\ldots LS \ldots \quad \rightarrow \quad \ldots SL \ldots, \tag{14}$$

which is shown explicitly in Fig.4. In this sense, phasons do not corresponds to oscillations of the atoms around the equilibrium position (like phonons) but rather correspond to the

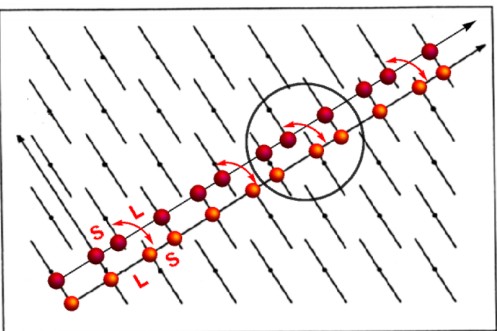 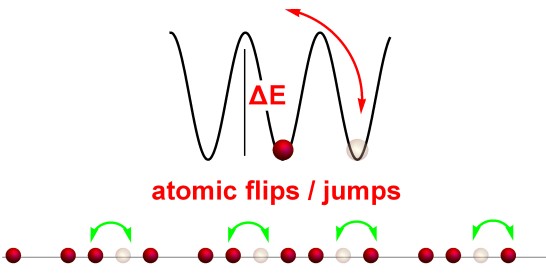

Figure 4: The equivalence between internal phason-displacements and atomic-flips/jumps. **Left:** The effects of the phason-displacement within the *superspace* description. The internal shift makes some atoms to flips and the quasicrystal structure to change locally from $LS \rightarrow SL$. **Right:** A zoom on the atomic flips induced by the phason-displacement. Notice that in order for the atoms to flips a energy potential $\Delta E$ has to be over passed. In this sense, atomic flips/jumps cost energy and they can happen only at finite temperature and in a diffusive fashion.

rearrangements of their positions. Such atomic re-arrangements have been observed using high resolution transmission electron microscopy (TEM) [68,69]. The corresponding dynamics are therefore diffusive and not propagative as that of standard phonons. In a sense, their motion is much more similar to solid diffusion and mass motion [70][7]. The phason mode is diffusive whenever the free energy is analytic in the gradients of the phason displacements [43].

From an energetic point of view, in the equilibrium configuration, every atom sits at the minimum of a potential. As a result, the flips induced by the phasons displacements correspond to jumps of energy barriers $\Delta E$ within such potential (see right panel of Fig.4). Fundamentally, this is why phason can exist only at finite temperature and their diffusion constant vanishes in the limit of $T \rightarrow 0$. At zero temperature, there is no available kinetic energy to overcome the potential energy barriers. This point is also related to the fact that the generator of infinitesimal shifts in the transverse direction—or, if you want, in the phase—does not commute with the Hamiltonian $H$. This is an important feature that we will recover in the EFT formulation.

Finally, it is worth noting that phason modes can be viewed as modes with a wavevector in the parallel space and a polarization in the perpendicular space [57]. Interestingly enough, phasons are thought to be essential for the stability of quasicrystals in the same way fleuxural phonons are for graphene [71].

Moreover, their diffusive nature at short wavelength is responsible for a peculiar linear in $T$ contribution to the specific heat at low temperature [72, 73]. This contribution is indeed typical of diffusive modes and it can be explained by means of a simple effective theory [74]. Interestingly, quasicrystals share a lot of anomalous properties with glasses and disordered systems [75].

Before going to the next section, let us comment on the role of the phason in modulated structures such as incommensurate superlattices. In 1D incommensurate structures it has been shown that anharmonicities can pin the sliding mode (analogous to the phason in quasicrys-

---

[7]In a closer sight, the diffusive phason dynamics differs from solid diffusion or mass motion because it does not arise from the conservation of a charge as standard diffusive modes (e.g. charge diffusion, shear diffusion, etc ...) .

tals) and remove it from the set of hydrodynamic modes. This process, known as the breaking of analiticity [76], can be studied by means of simple discrete and continuous models, the most famous one being the Frenkel-Kontorova model [77]. In that setup, which in the continuous limit reduces to the Sine-Gordon model [78], the fundamental dynamics is played by the modulation phase $\phi$. Its profile is given by a soliton solution which physically represents a domain wall between two commensurate phases, or in other words the extra-atom added to the commensurate chain. Depending on the strength of interactions, this mode gets pinned in the commensurate phase. On the contrary, this does not happen for the phason in quasicrystals. Thus, quasicrystals seem to be different from the other classes of quasiperiodic systems [59]. To the best of our knowledge, a satisfactory explanation is still absent.

### 1.3 A $k-$gap intermezzo

The hydrodynamic behavior of phasons modes in quasicrystals was analyzed in the seminal work [43]. The crucial point consists in adding to the hydrodynamics the phasons displacement $\vec{w}$ together with the standard phonons displacement $\vec{u}$. Dropping the various technicalities, we can write down the equation of motion for the displacement $w$.[8] A common practice is to consider the conjugate momentum $g_w \equiv \rho\, p_w$ where $\rho$ is the mass density. The equation of motion for the conjugate momentum reads [43] :

$$\partial_t\, g_w + \gamma\, g_w = -\frac{\delta\mathcal{F}}{\delta w}, \tag{15}$$

where $\mathcal{F}$ is the free energy. By considering the expansion of the free energy in terms of the gradients of the displacement, one could realize that:

$$\frac{\delta\mathcal{F}}{\delta w} \sim \nabla^2 w. \tag{16}$$

Finally, we can see that in Fourier space $g_w = \partial_t w = i\,\omega\,w$ such that the final equation of motion takes the form

$$\omega^2 + i\,\gamma\,\omega = v^2\,k^2, \tag{17}$$

which is one of the main results of [43]. This equation predicts that at small momenta the phason dynamics are diffusive

$$\omega = -i\,D\,k^2 \qquad \text{with} \qquad D = v^2/\gamma; \tag{18}$$

on the contrary, in the opposite regime, $\omega \gg \gamma$, the phason becomes a propagating sound mode

$$\omega = v\,k. \tag{19}$$

Notice that if $\gamma = 0$ then the diffusive dynamics disappears completely. The term proportional to $\gamma = \tau^{-1}$ describes a relaxation process and it is related to the fact that the phason displacements do not commute with the Hamiltonian. As we will see later in the EFT description, this means that the corresponding internal translation symmetry has no corresponding Noether current.[9] The phason dispersion relation is shown in the left panel of Fig.5 and it is known as $k-$gap. The dispersion relation has a gap in momentum space and it displays a crossover between a diffusive behaviour at large length-scale to a propagating one at small scales. This crossover has been studied in detail in the context of incommensurate structures [62,79–81]. The connection between the *telegraph equation* and the dynamics of phasons in quasicrystals

---

[8]For simplicity, we consider only one internal dimension; $w$ stands for the displacement along such direction.

[9]We will see that EFTs describing dissipative systems at finite temperature can possess symmetries without a corresponding non-trivial Noether current.

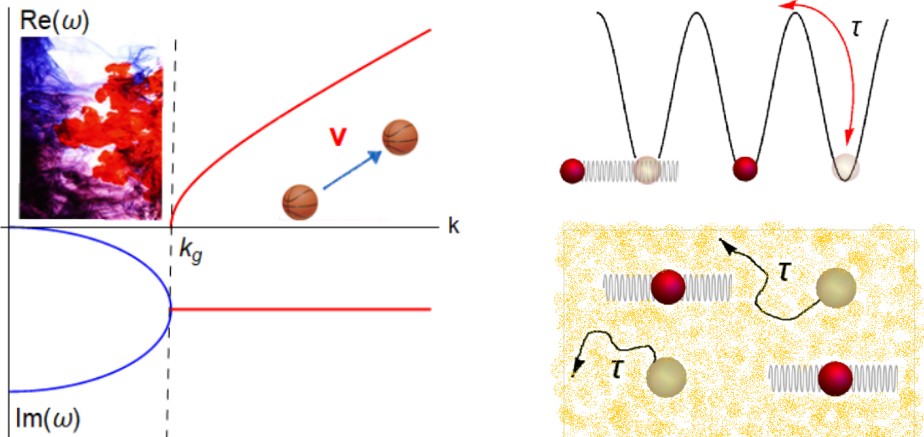

Figure 5: **Left:** The phason dispersion relation arising from the hydrodynamic description of quasicrystals [43]. The upper half plane is the real part and the lower the imaginary part. The blue color indicates the diffusive regime while the red one the propagating one. $k = k_g$ is the crossover point known as $k$−gap. **Right:** The dynamics from the energetic point of view and in the physical space. The jumps in the potential correspond to te re-arrangement in the physical space of the quasicrystal. The time $\tau$ is either the average jump time or the average time of molecular re-arrangements.

have been already discussed using elasto-dynamics in [82].

The effective relaxation rate $\gamma = \tau^{-1}$ is physically given by the fact that a phason displacement corresponds to a jump in the potential energy whose average time is given by $\tau$. These jumping processes are always accompanied by the small vibrations around the equilibrium configuration, which give rise to the r.h.s. of Eq.(17). In physical space, the term proportional to $\tau$ defines the re-arrangement of the atomic positions. The re-arrangements are possible only at finite temperature and are crucial for the diffusive nature of the phasons. Therefore zero-temperature field theories such as [83–86] miss this diffusive dynamics and are not able to compare successfully with the holographic results [34–41].

This scenario is not new and it can be found in several dissipative systems. A complete review on the topic can be found in [6]. The specific equation (17), which is sometimes known as *telegraph equation*, has been observed in several holographic systems [87–90] and recently described with formal field theory methods [91]. Interestingly enough, it is the typical behaviour found in the context of diffusive Goldstone bosons [31–33]. In this last scenario, the relaxation parameter $\tau$ arises because of the "open" nature of the system. For example, it directly appears in the Langevin equation as a friction term or from the coupling to an external bath in a open system.

Let us conclude this section by noticing that the phasons dynamics display striking similarities with that of the shear modes in liquids (see for example the molecular dynamical simulations of [92]). To the best of our knowledge, this parallelism has never been noticed before and it deserves further analysis.

### 1.4 A brief introduction to EFT methods

Most many-body systems like quasicrystals exist at finite temperature. As a result, the equilibrium state of the system takes the form of a thermal density matrix given by

$$\rho = \frac{e^{-\beta H}}{\mathrm{tr}(e^{-\beta H})}, \tag{20}$$

where $H$ is the Hamiltonian and $\beta \equiv 1/T$ is the inverse equilibrium temperature of the system. For finite values of the temperature, this equilibrium density matrix describes a mixed state. As a result, the usual quantum field theory techniques designed to compute S-matrix elements using in-out states and vacuum correlators are of no use. Instead, we must compute time-dependent correlators in the presence of a thermal density matrix. A well-known technique for doing so is the in-in formalism defined on the Schwinger-Keldysh (SK) contour [26]. In this formalism, the sources that appear in the generating functional are doubled. Let $U(+\infty, -\infty; J)$ be the unitary time-evolution operator in the presence of external source $J$. We define the SK generating functional by

$$e^{W[J_1, J_2]} \equiv \mathrm{tr}[U(+\infty, -\infty; J_1) \rho \, U^{\dagger}(+\infty, -\infty, J_2)]. \tag{21}$$

Notice that the inclusion of two sources is necessary to obtain a non-trivial generating functional; if $J_1 = J_2$, then the cyclicity of the trace ensures that $W$ vanishes.

In general, many-body systems have too many degrees of freedom to keep track of. We therefore would like a course-grained description of the degrees of freedom that persist over large distance and time scales. Essentially, we would like to construct an EFT for the infrared degrees of freedom, denoted by $\varphi$, that can reproduce the generating functional $W[J_1, J_2]$. Often since we are interested in the infrared degrees of freedom, $\varphi$ are just the Goldstone modes associated with spontaneous breaking of the Poincaré and internal symmetry groups. Because we have doubled sources, our effective action must have doubled field content, $\varphi_1$ and $\varphi_2$. Coupling to the sources $J_1$ and $J_2$, we define the non-equilibrium effective action $I_{\mathrm{EFT}}[\varphi_1, \varphi_2; J_1, J_2]$ such that

$$\int_{\mathrm{SK}} \mathcal{D}[\varphi_1 \varphi_2] e^{iI_{\mathrm{EFT}}[\varphi_1, \varphi_2; J_1, J_2]} = e^{W[J_1, J_2]}, \tag{22}$$

where the subscript SK indicates that we impose the SK boundary conditions, namely that in the distance future the two copies of the fields agree $\varphi_1(+\infty) = \varphi_2(+\infty)$. This future-time boundary condition is a remnant of the trace in (21) and has a very important consequence: even though the field content is doubled, the global symmetry content is not. Thus there is just one copy of the global symmetry group.

Often, it is convenient to work in the so-called retarded-advanced basis defined by symmetric and anti-symmetric combinations of the fields. We have

$$\varphi_r \equiv \frac{1}{2}(\varphi_1 + \varphi_2), \qquad \varphi_a \equiv \varphi_1 - \varphi_2. \tag{23}$$

It turns out that these two fields play very different roles: $\varphi_r$ plays the role of a classical field (or expectation value of a quantum field), while $\varphi_a$ encodes information about thermal and quantum fluctuations about the classical solution.

The program of non-equilibrium EFT bears a close resemblance to ordinary EFT in that we use symmetry as the guiding principle. However, there are some additional subtleties that arise in non-equilibrium EFT. We enumerate the rules for constructing such effective actions below without proof (see [93] for a review).

- Unlike ordinary actions, $I_{\text{EFT}}$ may be complex. There are three important constraints that come from unitarity, namely

$$
\begin{aligned}
I_{\text{EFT}}[\varphi_1, \varphi_2; J_1, J_2] &= -I_{\text{EFT}}^*[\varphi_2, \varphi_1; J_2, J_1] \\
\text{Im} I_{\text{EFT}}[\varphi_1, \varphi_2; J_1, J_2] &\geq 0 \text{ for any } \varphi_{1,2}, J_{1,2} \\
I_{\text{EFT}}^*[\varphi_1 = \varphi_2; J_1 = J_2] &= 0.
\end{aligned}
\tag{24}
$$

- Any symmetry of the ultraviolet theory is also a symmetry of $I_{\text{EFT}}$ except for symmetries that reverse the direction of time. The fact the time-reversing transformations are not symmetries of the infrared theory allows the production of entropy and therefore the introduction of dissipation.

- If the equilibrium density matrix $\rho$ takes the form of a thermal density matrix, then the effective action is invariant under the so-called dynamical KMS symmetries. Suppose that the ultraviolet theory is invariant under the time-reversing anti-unitary symmetry transformation $\Theta$. Then, setting the sources to zero, we have that the infrared theory enjoys the symmetry transformations

$$
\begin{aligned}
\varphi_1(x) &\to \Theta\, \varphi_1(t - i\,\theta, \vec{x}), \\
\varphi_2(x) &\to \Theta\, \phi_2(t + i\,(\beta_0 - \theta), \vec{x}),
\end{aligned}
\tag{25}
$$

for any $\theta \in [0, \beta_0)$. It is straightforward to check that these transformations are their own inverse. To take the classical limit is is convenient to work in the basis (23). Then the classical dynamical KMS symmetries become

$$
\begin{aligned}
\varphi_r(x) &\to \Theta\, \varphi_r(x), \\
\varphi_a(x) &\to \Theta\,[\varphi_a(x) + i\,\beta_0\,\partial_t \varphi_r(x)].
\end{aligned}
\tag{26}
$$

Notice that the change in $\varphi_a$ is related to the time derivative of $\varphi_r$. Thus, to have a consistent derivative expansion, we must count $\partial_t \varphi_r$ and $\varphi_a$ at the same order in the derivative expansion.

As a quick warm-up we now consider the construction of the non-equilibrium EFT describing a system in fluid phase such that the only conserved quantity is the stress-energy tensor. It turns out that the construction of the fluid EFT is most conveniently performed on a manifold other than the physical spacetime, which we will call the *fluid manifold*. The coordinates of the fluid manifold are $\phi^M$ for $M = 0, 1, 2, 3$. We expect that the infrared degrees of freedom are related to conserved quantities because all other quantities can locally relax to equilibrium. Thus, since only the stress-energy tensor is conserved, the relevant sources in the generating functional $W$ are the spacetime metric tensors $g_{1\mu\nu}$ and $g_{2\mu\nu}$. In this way differentiating with respect to these metrics yields correlation functions among the stress-energy tensor. Thus, the generating functional becomes

$$
e^{W[g_{1\mu\nu}, g_{2\mu\nu}]} \equiv \text{tr}[U(+\infty, -\infty; g_{1\mu\nu})\, \rho\, U^\dagger(+\infty, -\infty, g_{2\mu\nu})].
\tag{27}
$$

Since the stress-energy tensor is covariantly conserved, $W$ must be invariant under two independent diffeomorphism symmetries. Let $X_s^\mu(\phi)$ for $s = 1, 2$ be two coordinate transformations. Defining the pull-back metrics $G_{sMN} \equiv \frac{\partial X_s^\mu}{\partial \phi^M} g_{s\mu\nu}(X_s(\phi)) \frac{\partial X_s^\nu}{\partial \phi^N}$, we have that

$$
W[g_{1\mu\nu}, g_{2\mu\nu}] = W[G_{1MN}, G_{2MN}].
\tag{28}
$$

Now we use the Stückelberg trick and promote these coordinate transformations to dynamical fields $X_s^\mu(\phi)$, which we can think of a embedding the fluid worldvolume into the physical

Physical spacetime (2)          Fluid worldvolume          Physical spacetime (1)

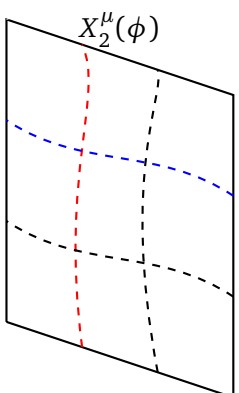    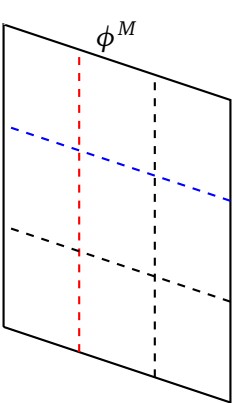    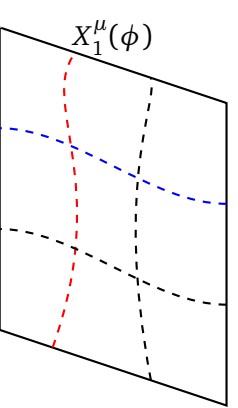

Figure 6: This figure depicts how the fluid world-volume with coordinates $\phi^M$ is mapped into two copies of the physical spacetime by the maps $X_1^\mu(\phi)$ and $X_2^\mu(\phi)$. The red and blue lines in the right-and left-hand panels are the images of the red and blue lines in the middle panel under the maps $X_1^\mu(\phi)$ and $X_2^\mu(\phi)$.

spacetime. In particular, we 'integrate in' the fields $X_s^\mu(\phi)$ such that the generating functional becomes

$$e^{W[g_{1\mu\nu}, g_{2\mu\nu}]} = \int \mathcal{D}[X_1 X_2] e^{iI_{\text{EFT}}[G_{1MN}, G_{2MN}]}. \tag{29}$$

We thus have our non-equilibrium effective action $I_{\text{EFT}}[G_{1MN}, G_{2MN}]$. However, at this point, our action is not guaranteed to describe a system in fluid phase. To describe a system in fluid phase, it turns out that we must impost the partial diffeomorphism gauge symmetries

$$\phi^0 \to \phi^0 + f(\phi^i), \qquad \phi^i \to g^i(\phi^j), \tag{30}$$

where $f$ and $g^i$ are generic functions of the spatial coordinates $\phi^i$ for $i = 1, 2, 3$. At leading order in the derivative expansion, the EFT constructed with the symmetries (30) is physically equivalent to the standard EFT with volume preserving diffeomorphisms typical of fluids [12]; see [42] for more details. A more complete discussion of the fluid EFT can be found in [42, 94, 95].

## 2 The EFT construction for quasicrystals

We now turn our attention to the construction of an EFT for quasicrystals. Like fluids, quasicrystals exist at finite temperature and enjoy conservation of the stress-energy tensor. We will therefore construct our effective action on the fluid worldvolume, whose coordinates, $\phi^M$, enjoy the gauge symmetry (30) and we will have the embedding coordinate fields $X_s^\mu(\phi)$.

Quasicrystals are most conveniently conceived of in terms of an embedding into a higher-dimensional space, which we term the quasicrystal space, with coordinates $\psi^A$ for $A = 1, 2, 3, 4$ [10]. At the level of EFT, we introduce the fields $\psi_s^A(\phi)$, which embed the fluid worldvolume into the higher-dimensional quasicrystal space.[11] If we are concerned with dynamics on distance scales much larger than the atomic spacing, we expect our theory to be invariant under

---

[10]As explained in the introduction, the dimension of the space spanned by the $A$ index parametrizes the rank of the quasicrystal. For simplicity, we consider only the situation with $A = 1, 2, 3, 4$ (e.g. rank $D - d = 1$) .

[11]Notice that unlike the fluid worldvolume, the quasicrystal space has no time-like coordinate.

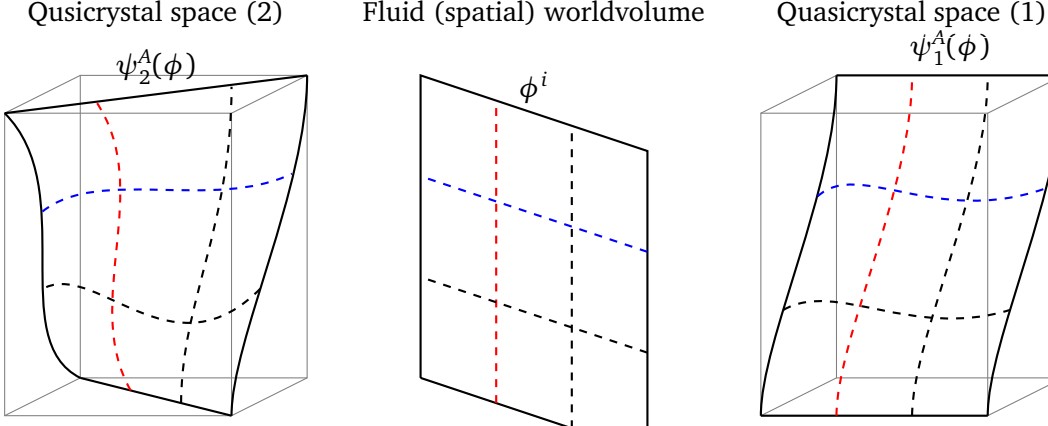

Figure 7: This figure depicts how, at fixed time $\phi^0$, the spatial submanifold of the fluid worldvolume is mappeed into two copies of the quasicrystal worldvolume with the embedding maps $\psi_1^A(\phi)$ and $\psi_1^B(\phi)$. We suppress the $\phi^0$-coordinate as the quasicrystal spaces have no intrinsic notion of time. Notice that unlike the mapping into physical spacetime, the quasicrystal dimension is greater than that of the spatial fluid worldvolume.

continuous translations in the quasicrystal space. We thus impose the shift symmetries

$$\psi_s^A \to \psi_s^A + \lambda^A, \tag{31}$$

for constants $\lambda^A$. Notice that because our EFT is defined on the SK contour, in the distant future $\psi_1^A(+\infty) = \psi_2^A(+\infty)$. As a result, there can be only one copy of the above shift symmetries despite the fact that the field content is doubled. The coordinates $\psi^A$ represent coordinates of a four-dimensional lattice. Such lattices often enjoy certain rotational symmetries. Suppose these rotational symmetries form the discrete subgroup $\mathcal{S} \subset SO(4)$, then we expect the EFT to be invariant under the transformations

$$\psi_s^A \to \mathcal{O}^{AB}\psi_s^B, \qquad \mathcal{O} \in \mathcal{S}. \tag{32}$$

We are now in a position to construct the quasicrystal EFT. Let us begin by constructing covariant terms. We are currently only interested in the classical EFT. Taking the classical limit is most conveniently performed in the retarded-advanced basis. Thus, we will construct covariant terms using the fields

$$X_r^\mu \equiv \frac{1}{2}(X_1^\mu + X_2^\mu), \quad X_a^\mu \equiv X_1^\mu - X_2^\mu, \quad \psi_r^A \equiv \frac{1}{2}(\psi_1^A + \psi_2^A), \quad \psi_a^A = \psi_1^A - \psi_2^A. \tag{33}$$

For ease of notation, define the matrix $e_M^\mu \equiv \partial_M X_r^\mu$. Then, in the classical limit, we have

$$\beta^\mu \equiv \beta_0\, e_0^\mu, \qquad \partial_\mu X_a^\nu \equiv (e^{-1})_\mu^M\, \partial_M X_a^\nu, \tag{34}$$

where $\beta_0$ is the equilibrium inverse temperature. It turns out that $\beta^\mu$ can be interpreted as the local inverse temperature four-vector. Additionally, we have convariant building blocks constructed using the quasicrystal fields. Notice that (31) leaves $\psi_a^A$ invariant while shifting $\psi_r^A$ by a constant. Therefore,

$$\partial_\mu \psi_r^A \equiv (e^{-1})_\mu^M\, \partial_M \psi_r^A, \qquad \psi_a^A, \tag{35}$$

are covariant building-blocks of our EFT.

The last consideration before we can construct the leading-order effective action is to determine how the fields transform under the classical dynamical KMS symmetries. The spacetime embedding fields transform by

$$X_r^\mu \to \Theta X_r^\mu, \qquad X_a^\mu \to \Theta X_a^\mu - i\,\Theta\,\beta^\mu + i\,\beta_0\,\delta_0^\mu, \tag{36}$$

and the quasicrysal fields transform by

$$\psi_r^A \to \Theta\,\psi_r^A, \qquad \psi_a^A \to \Theta\,\psi_a^A - i\,\Theta\,\beta_0\frac{\partial\psi_r^A}{\partial\phi^0}, \tag{37}$$

where $\Theta$ is a time-reversing symmetry transformation of the ultraviolet theory.[12] It will be convenient to define the local inverse temperature scalar $\beta \equiv \sqrt{-\beta^\mu\beta_\mu}$, the fluid four-velocity $u^\mu \equiv \beta^\mu/\beta$, the symmetric matrix $Y^{AB} \equiv \partial_\mu\psi_r^A\partial^\mu\psi_r^B$, and the column vector (in quasicrystal space) $Z^A \equiv \beta^\mu\partial_\mu\psi_r^A$.

In the derivative expansion, $\beta^\mu$ and $\partial_\mu\psi^i$ for $i=1,2,3$ count as zeroth order because their equilibrium expectation values are non-vanishing constants. Therefore the effective action may depend on arbitrary Poincaré-invariant functions of these building-blocks. The phason building-block $\partial_\mu\psi_r^4$, however, has vanishing expectation value. We therefore count it as first order in derivatives. Similarly, we count $\psi_a^A$ and $X_a^\mu$, which all have vanishing expectation value, at first order in derivatives. We therefore count $\partial_\mu\psi_a^A$ and $\partial_\mu X_a^\nu$ at second order in derivatives. Thus, performing a coodinate transformation so that our EFT is defined on the physical spacetime $x^\mu \equiv X_r^\mu$, the leading order Lagrangian with non-trivial dynamics is

$$\mathcal{L}_{\text{EFT}} = T^{\mu\nu}\partial_\mu X_{a\nu} + J^{A\mu}\partial_\mu\psi_a^A + \Gamma^A\psi_a^A + \frac{i}{2}M^{AB}\psi_a^A\psi_a^B, \tag{38}$$

where

$$T^{\mu\nu} = \epsilon(\beta, Y^{AB}, Z^A)u^\mu u^\nu + p(\beta, Y^{AB}, Z^A)\Delta^{\mu\nu} + r^{AB}(\beta, Y^{AB}, Z^A)\partial^\mu\psi_r^A\partial^\nu\psi_r^B, \tag{39}$$

is the stress-energy tensor such that $\Delta^{\mu\nu} = \eta^{\mu\nu} + u^\mu u^\nu$,

$$J^{A\mu} = F^A(\beta, Y^{AB}, Z^A)u^\mu + H^{AB}(\beta, Y^{AB}, Z^A)\partial^\mu\psi_r^B, \tag{40}$$

and $\Gamma^A$ and $M^{AB}$ are generic functions of $\beta$, $Y^{AB}$, and $Z^A$. Notice that non-derivative terms in the $\psi_r$ field (but not in the $\psi_a$ one) are forbidden because of the shift symmetry (31). Finally, one could include a conserved $U(1)$ charge associated with particle-number conservation. This will lead to a diffusive mode (e.g. mass diffusion or electric charge diffusion). In the interest of simplicity, we will not include this $U(1)$-mode in our theory, but interested readers can consult [42] to understand how to implement it. Importantly, the diffusive behavior of the phason has nothing to do with the conservation of this charge or any other. In fact, as we will see, it arises precisely because there is no conserved Noether charge associated with the quasicrystal shift symmetry $\psi_a^4 \to \psi_a^4 + \lambda^4$.

The various coefficient functions, $\epsilon$, $p$, $r^{AB}$, $F^A$, $H^{AB}$, $\Gamma^A$, and $M^{AB}$, however are not totally independent on one another as we have not yet imposed the (classical) dynamical KMS symmetry. It is straightforward (though tedious) to show that the dynamical KMS symmetry imposes the following relations:

$$\epsilon + p = -\beta\frac{\partial p}{\partial\beta}, \quad r^{AB} = \frac{\partial p}{\partial Y^{AB}}, \quad F^A = \frac{\partial p}{\partial Z^A}, \quad H^{AB} = 2\frac{\partial p}{\partial Y^{AB}}, \quad \Gamma^A = -M^{AB}Z^B. \tag{41}$$

---

[12]The dynamical KMS transformation of $X_a^0$ is somewhat unusual; see [94,95] for more information.

The first equation is nothing else that the standard thermodynamic law $\epsilon + p = sT$. With the relations (41), this Lagrangian (38) is the most generic EFT at leading order in the derivative expansion that is consistent with symmetries.

The effective action in Eq.(38) is the leading order result where higher-derivative corrections are neglected. Those corrections would modify the dispersion relations of the collective modes only at higher frequencies/momenta $\mathcal{O}(k^4), \mathcal{O}(\omega^4), etc$ and they could be systematically introduced if necessary. Notice also that our initial effective action 38 is Poincaré invariant. This assumption can be in principle (but not necessarily) be relaxed to non-relativistic symmetries such as Galilean invariance.

## 2.1 Noether's theorem

In ordinary field theory, the relationship between symmetries and conserved quantities is very straightforward: Noether's theorem guarantees that there is a one-to-one correspondence among symmetries and conserved quantities. In non-equilibrium systems, like the quasicrystal; however, such a straightforward relationship no longer exists [96, 97]. To see that this is so, consider the shift symmetries in (31). Are there conserved quantities associated with them? If there are, this is quite bad as the only fundamental continuous symmetries that exist in the ultraviolet theory are the Poincaré symmetries. As a result, the only conserved quantities should be the stress-energy tensor $T^{\mu\nu}$.

Ignoring the physical interpretation of our theory, since the Lagrangian (38) possesses the shift symmetries (31), Noether's theorem guarantees that the existence of currents that are conserved when the the equations of motion are satisfied. In particular, they are given by

$$K^{A\mu} = \frac{\partial \mathcal{L}_{\text{EFT}}}{\partial(\partial_\mu \psi_r^A)}. \tag{42}$$

However, notice that (I) every term in $\mathcal{L}_{\text{EFT}}$ and hence every term in $K^{A\mu}$ contains at least one $a$-type field and (II) when the equations of motion are satisfied, all $a$-type fields vanish. Thus, when the equations of motion are satisfied, $K^{A\mu}$ vanishes. Thus, it is unphysical and the fact that it is conserved contains no physical content. In this way, Noether's theorem can be satisfied at the level of mathematics, while containing no physical significance. Other examples of non-equilibrium EFTs that enjoy shift symmetries and no corresponding Noether current can be found in [96,97].

Now we contrast this result with the conservation of the stress-energy tensor. Notice that the equations of motion for $X_a^\mu$ give

$$\partial_\nu T^{\mu\nu} = 0. \tag{43}$$

Thus, the stress-energy tensor is conserved, as desired. The fact that $T^{\mu\nu}$ is conserved, however, is *not* a consequence of Poincaré symmetry. To see that this is so, notice that $X_a^\mu$ is a Lorentz covariant object.[13] As such, it is consistent with Poincaré symmetry to include a term of the form $\gamma^\mu X_a^\mu$, where $\gamma^\mu$ is some Lorentz four-vector constructed from $r-type$ fields. The inclusion of such a term would yield the equations of motion

$$\partial_\nu T^{\mu\nu} = \gamma^\mu, \tag{44}$$

thereby killing the conservation of energy and momentum. The way we forbid the inclusion of $X_a^\mu$ without derivatives is by gauging spacetime translation symmetry. Recall that in the construction of the fluid EFT, we had to introduce two metric tensors $g_{1\mu\nu}$ and $g_{2\mu\nu}$. Because

---

[13]Recall that there is only one copy of the global symmetry group because of the future-time SK boundary conditions. As a result, $X_a^\mu$ is translation-invariant.

gauge symmetries (in this case diffeomorphisms) are local in spacetime, we are allowed to have two copies of them; that is, the future-time SK boundary condition, $X_1^\mu(+\infty) = X_2^\mu(+\infty)$, permits two independent diffeomorphism symmetries as long as they vanish in the distant future. With this double copy of diffeomorphism gauge symmetry, $X_s^\mu$ become Stückleberg fields and can only appear in the forms

$$G_{sMN}(\phi) = \frac{\partial X_s^\mu(\phi)}{\partial \phi^M} g_{s\mu\nu}(X_s(\phi)) \frac{\partial X_s^\nu(\phi)}{\partial \phi^N}. \tag{45}$$

Then, to recover the Lagrangian (38), we ultimately remove the metric sources by fixing $g_{s\mu\nu} = \eta_{\mu\nu}$. Thus, $X_a^\mu$ is not a permitted building-block and the stress-energy tensor is conserved.

Finally, it is worth commenting on what happens if we allow a non-trivially conserved current to exist that corresponds to translations in quasicrystal space. This can be accomplished by introducing external source $U(1)$ gauge fields corresponding to the shift symmetries (31) and then setting these source fields to zero at the end. This will have the effect of setting $\Gamma^A = 0$, meaning that the equations of motion for $\psi_a^A$ yield

$$\partial_\mu J^{A\mu} = 0. \tag{46}$$

The conservation of $J^{i\mu}$ for $i = 1, 2, 3$ (i.e. the currents associated with phonons) leads to the phenomenon of so-called "second sound" in the limit of zero Umklapp scattering [97]. This is to say that in addition to the usual transverse and longitudinal sound waves that exist in solids and quasicrystals, there is an additional hydrodynamic sound mode that propagates independently. When $J^{i\mu}$ are not conserved, this second sound mode becomes diffusive at low momentum; $\Gamma^i$ sets the momentum scale below which diffusion occurs. If we are only interested in momentum states below this momentum scale, then we may integrate out $\psi_{r/a}^i$. At leading order in the derivative expansion, the equations of motion for $\psi_{r/a}^i$ yield

$$Z^i = 0, \qquad \psi_a^i = 0. \tag{47}$$

The first equation can be solved to give $\psi_r^i = \alpha^i(\phi^j)$, for arbitrary spatial functions $\alpha^i$. Using the partial diffeomorphism symmetry of the fluid worldvolume (30), we may gauge-fix $\alpha^i = \phi^i$, that is $\psi_r^i = \phi^i$. After integrating out the solid phonon fields $\psi_{r/a}^i$ and gauge-fixing, we find that "fluid worldvolume" has residual diffeomorphism symmetry

$$\phi^0 \to \phi^0 + f(\phi^i), \qquad \phi^i \to \phi^i + \lambda^i, \tag{48}$$

where $f$ is a generic function of the spatial coordinates and $\lambda^i$ are constants. Since our worldvolume has reduced symmetry, it no longer corresponds to a fluid in any way; we will therefore re-name it the "solid worldvolume". Now the remaining fields are

$$X_r^\mu(\phi), \qquad X_a^\mu(\phi), \qquad \psi_r^4(\phi), \qquad \psi_a^4(\phi). \tag{49}$$

The first two fields describe the embedding into the physical spacetime and the last two describe fluctuations in the quasicrystal space perpendicular to the solid worldvolume and hence correspond to the phason degrees of freedom.[14]

Since we have integrated out $\psi_{r/a}^i$, we lose some building blocks, namely $Z^i$, and $\psi_a^i$, whereas other are altered like $Y^{AB}$. In particular we find that $Y^{AB}$ is replaced by $y^{AB}$, where[15]

$$y^{ij} = (e^{-1})_\mu^i \eta^{\mu\nu}(e^{-1})_\nu^j, \qquad y^{4i} = y^{i4} = (e^{-1})_\mu^i \partial^\mu \psi_r^4, \qquad y^{44} = \partial_\mu \psi_r^4 \partial^\mu \psi_r^4. \tag{50}$$

Thus, our new effective Lagrangian is identical to (38) except with the replacements

$$Z^i \to 0, \qquad \psi_a^i \to 0, \qquad Y^{AB} \to y^{AB}. \tag{51}$$

---

[14]Notice that we are assuming the quasicrystal space has one additional dimension beyond the physical spatial dimensions for the sake of simplicity.

[15]Recall that $e_M^\mu \equiv \partial X_r^\mu / \partial \phi^M$.

## 2.2 Phasons from symmetries with no Noether currents

It is often claimed that at low momentum, phonons propagate via waves whereas phasons are diffusive. The arguments for why phasons should be diffusive, however, are often quite vague or depend entirely on empirical observations. Here we will see that the low-momentum behavior of phasons given by our non-equilibrium EFT are diffusive. We will then recover the empirically verified result that above a certain momentum, phasons yield propagating waves.

The quickest path to understand the dispersion relation of phasons is as follows. Consider the Lagrangian (38) and suppose that only the phason is excited, that is $\psi^4$ is free to fluctuate, while $X^\mu$ and $\psi^i$ remain locked in place. Then the linearized equation of motion for $\psi_a^4$ is

$$H^{44} \partial_\mu \partial^\mu \psi_r^4 + \frac{\partial F^4}{\partial Z^4} \partial_0^2 \psi_r^4 - M^{44} \partial_0 \psi_r^4 = 0. \tag{52}$$

Transforming to Fourier space, we have $\partial_0 \to -i\omega$ and $\partial_i \to k_i$, yielding

$$\omega^2 + i\gamma\omega = v^2 k^2, \tag{53}$$

where

$$\gamma \equiv \frac{M^{44}}{H^{44} - \partial F^4/\partial Z^4}, \qquad v^2 = \frac{H^{44}}{H^{44} - \partial F^4/\partial Z^4}. \tag{54}$$

Thus, at low momentum the phason is diffusive, whereas at high momentum it yields a propagating wave. Notice that the $M^{44}$ term, producing the finite relaxation time $\sim \gamma^{-1}$ comes from the non-hermitian part of the action (38) and it is therefore directly connected to dissipation[16]. On the contrary, the $H^{44}$ term can be considered as an additional elastic modulus coming from phason elasticity, as in the hydrodynamic treatment of [43].

We can see that the diffusive behavior follows directly from the fact that $J^{4\mu}$ is not conserved in the following sense. In the previous subsection, we saw how even though the Lagrangian is invariant under the quasicrystal shift symmetries, there need not be a corresponding conserved current as long as $\psi_a^A$ is an allowed building-block. However, if we require that $\psi_a$ always appear with derivatives, then we do have a conserved current, namely $J^{A\mu}$. Suppose for the sake of argument, we forbid the use of $\psi_a^4$ as a building block of the EFT such that $J^{4\mu}$ became conserved. Then we would have to fix $M^{44} = \Gamma^4 = 0$. This yields $\gamma = 0$, so the phonon dispersion relation becomes $\omega^2 = v^2 k^2$, that is it is a propagating wave with no gap. Thus, the fact that at low momentum the phason is diffusive is a direct result of the absent Noether current associated with the shift symmetry $\psi_s^4 \to \psi_s^4 + \lambda^4$. Just by computing the equations of motion from the action (38), we obtain:

$$\partial_\mu J^{4\mu} = -\Gamma^4 \sim M^{44}, \tag{55}$$

which confirms explicitly the previous arguments.

An equivalent statement often mentioned in the literature, is that the phason shifts leave the free energy unchanged but they do not commute with the hamiltonian of the system [62]. In particular, letting $\mathcal{P}_4$ be the generator of the quasicrystal shift symmetry, we expect that

$$\langle [H, \mathcal{P}_4] \rangle \sim \langle H^\dagger \mathcal{P}_4 - \mathcal{P}_4 H \rangle \sim \frac{d\langle \mathcal{P}_4 \rangle}{dt} \sim M^{44}, \tag{56}$$

which follows immediately from the non-Hermitian property of the last term in the action (38). Again, notice how this last non-Hermitian term, which is the key for the diffusive nature of the phason and the full story, makes sense only at finite temperature/dissipation.

---

[16]See [91] for another field theory with this property.

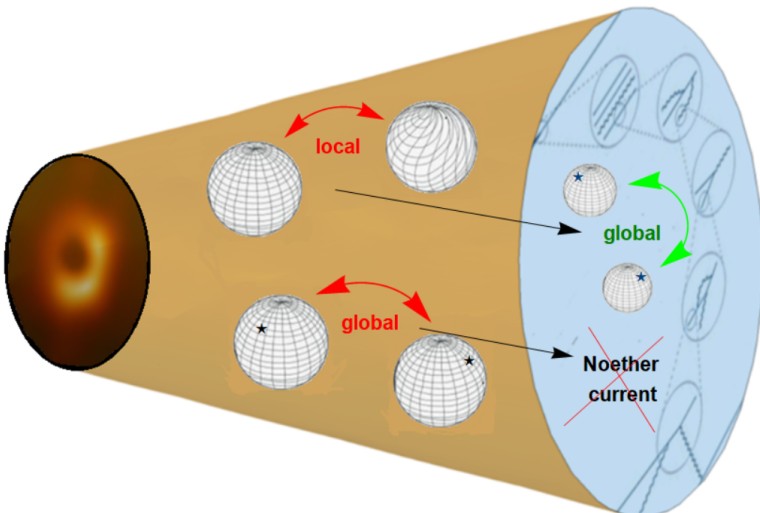

Figure 8: The holographic picture. Local (gauge) symmetries in the bulk map into global symmetries in the boundary field theory. Global symmetries in the bulk (e.g. a global rotation) correspond to symmetries without Noether currents/charges in the boundary field theory.

In the above discussion, we have neglected interactions between the phonon and phason degrees of freedom. These terms merely serve to complicate matters and do not change the form of the the phason dispersion relation. However, they will change the numerical values of $\tau$ and $\nu$ in the dispersion relation, so if one wishes to make precise predictions, these interactions ought to be considered. Such interactions can be included by considering all terms from (38) and are systematically treated in the hydrodynamic description of [43].

## 2.3 A brief comparison with holographic models

In recent years, in the holographic community (see [25] for a review), there has been fervent activity in understanding and modelling the spontaneous breaking of translational invariance and the corresponding elastic properties of the dual field theories, started with the seminal work [98]. Essentially for the sake of simplicity, a lot of attention has been devoted to the so-called *homogeneous models* such as axion-like models [99,100] and Q-lattice models [101].

In these models, spacetime translations $\mathfrak{P}$: $x^i \to x^i + a^i$ are broken together with an internal and global shift symmetry $\mathfrak{S}$: $\phi^I \to \phi^I + b^I$ to the diagonal subgroup $\mathfrak{P} \times_{diag} \mathfrak{S}$. Thanks to this symmetry-breaking pattern, the matter stress tensor and the resulting geometry do not display any coordinate-dependence and thus are perfectly homogeneous (making the computations much simpler). There are a number of interesting phenomena that emerge in this model. In particular, a new diffusive Goldstone boson has been observed in the longitudinal spectrum of the excitations [35, 36, 38]. Moreover, it has been explicitly proven that the existence of such a mode is a direct and unique consequence of the existence of this broken global symmetry [34, 41] living in the extra-dimensional bulk.

Although the dispersion relation of this mode can be captured by an appropriate hydrodynamic description, its fundamental nature remains elusive. Only recently, [37] has revealed several striking similarities between the homogeneous holographic models and the physics of quasicrystals, identifying this additional excitation as the diffusive phason mode.

We are now in the position to understand the holographic results and connect them with the finite-temperature EFT description. The key-point in our discussion revolves around the role of the internal shift symmetry (31). Similar such symmetries exist in the standard EFTs

for ideal fluids and zero-temperature solids [12]; however (and very importantly) our holographic models do not implement the same number of symmetries in the dual field theory side exactly because the internal shift symmetry is not gauged in the bulk [102]. In other words, they do not exactly represent the gravity duals for the field theories of [12]. The point that this internal symmetry is global in the bulk reflects in the fact that the dual symmetry is a symmetry of the dual field theory but it does not have any associated Noether current nor conserved Noether charge (see Fig.8). It was suggested in [103] that symmetries like this one have to be considered as outer automorphisms of the operator algebra of the dual field theory. This point deserves further investigation.

In any case, the global nature of the bulk symmetry and the absence of an associated Noether current are fundamental in determining the low momentum diffusive nature of the phason mode. Several comments are in order.

- The dynamics which we are discussing and which appear in these specific holographic models are strictly realized only at zero temperature. In the holographic scenario, at $T = 0$ there is no fingerprint left of the global symmetry in the bulk. The phason is totally frozen, in a way similar to the transverse phonons in a liquid at small wavevector. The same result is obtained from the EFT under investigation.

- As already emphasized, the phason shift symmetry is not associated to any Noether charge nor any conserved current. The resulting diffusive mode is a new type of Goldstone and not the standard result from a conservation equation + Fick's constitutive relation. The physical implication is that the diffusive mode under investigation does not correspond to the standard mass diffusion in solids as envisaged for example in the standard hydrodynamic treatment [104]. Also, it is quite unclear to us how the phasons dynamics and in particular the phasons jumps can be interpreted as some kind of defects motion[17].

- The way to restore the existence of a Noether current and an associated conserved charge is by mean of gauging the phason shifts. Interestingly, this has been already discussed and realized in holographic in [102]. Once the symmetry is gauged, we do expect the phason to become a fully propagating mode. In other terms, the gauging of the symmetry coincide with setting $M^{44}$ and consequently $\gamma$ to zero. It would be interesting to understand if this is what happens in other aperiodic crystals such as modulated lattices, where the phason is indeed propagating and not diffusive.

## 2.4 Restoring periodicity: incommensurate-commensurate transitions

The phason field $\psi_s^4$ is invariant under a shift symmetry $\psi_s^4 \to \psi_s^4 + \lambda^4$. This symmetry indicates that the free energy describing the quasicrystal is left unchanged by the atomic rearrangements associated with shifting $\psi_s^4$ by a constant. These atomic rearrangements do not cost any energy as long as the worldvolume associated with the quasicrystal slices through the higher-dimensional quasicrystal space at an incommensurate angle. Suppose now that the angle is commensurate, i.e. is a rational number times $2\pi$. Then, the resulting equilibrium configuration should be periodic, i.e. we now have a crystalline solid as opposed to a quasicrystal. Suppose further that the rational number that multiplies $2\pi$ is the ratio of two co-prime integers that are much greater than one. Then, the angle is 'almost incommensurate' in the sense that it should be very difficult to distinguish from a truly incommensurate angle.

---

[17]In incommensurate structures, this possibility could be related to the dynamics of the soliton degree of freedom close to commensurability (e.g. Frenkel Kontorova model). In such limit, the incommensurate structure can be described by a commensurate one plus a single dynamical and mobile solitonic defect.

Physically, this means that, although the atoms exhibit a repeating pattern, the periodicity of the repetition is very large compared with the atomic spacing. Thus, shifting $\psi_s^4$ should cost some free energy, but not very much. In the limit the angle becomes incommensurate, the periodicity of the lattice goes to infinity (that it is loses its periodicity) and the shift of $\psi_s^4$ by a constant leaves the free energy invariant. As a result, when there is a (large) periodic structure to the atomic positions, we expect

$$\psi_s^4 \to \psi_s^4 + \lambda^4 \tag{57}$$

be an *approximate* symmetry. Thus, $\psi_s^4$ has a small gap, whose smallness is protected by this approximate symmetry. As a result, the EFT may have a weak dependence on $\psi_r^4$ without derivatives.

With this shift symmetry explicitly broken, the effective action looks identical to (38) except that the coefficient function $\epsilon$, $p$, $r^{AB}$, $F^A$, $H^{AB}$, $\Gamma^A$, and $M^{AB}$ may now freely depend on $\psi_r^4$.

From a phenomenological point of view, the loss of periodicity can be viewed as an incommensurate-commensurate transition in which the phason is indeed known to acquire a finite gap and disappear from the hydrodynamic spectrum. In the commensurate phase, the structure becomes fully periodic with a single well-defined lattice wave-vector and the phason is not free anymore to slide around.

The physics of commensurate-incommensurate phase transitions is very rich [105] and to the best of our knowledge has been never tackled with modern Schwinger-Keldysh EFT techniques. We plan to consider this problem in the near future.

## 2.5 A universal relation between pinning frequency, damping and phason diffusion constant

In this subsection, we consider a slightly different (and simpler) scenario in which both translations and the phase shift symmetry are softly broken by an external source. The typical example is given by the role of impurities in incommensurate charge density waves [106]. Impurities prevent the free sliding of the phason mode which gets pinned. In other words, phase shifts now cost energy and they are no longer associated to a hydrodynamic mode.

Let us start from the EFT action (38), which is invariant under the shift symmetry $\psi^A \to \psi^A + \lambda^A$ together with standard spacetime translations $x^\mu \to x^\mu + c^\mu$. Under the shift symmetry, $\psi_a^A$ does not transform but $\psi_r^A$ shifts by a constant. Likewise, by performing a spacetime translation, $X_a^\mu$ does not transform but $X_r^\mu$ does. Therefore, to break the symmetries explicitly it is sufficient to add terms which are not derivatives in $\psi_{r/a}^A, X_{r/a}^\mu$. At leading order, the most relevant terms we can write are

$$\mathcal{L}_{breaking} = \mathfrak{r}^{\mu\nu} X_{r\mu} X_{a\nu} + \rho^{AB} \psi_r^A \psi_a^B - s^{\mu\nu} \dot{X}_{r\mu} X_{a\nu} - \sigma^{AB} \dot{\psi}_r^A \psi_a^B + \dots, \tag{58}$$

which can be further simplified, by assuming the simplest tensorial structure, into

$$\mathcal{L}_{breaking} = \omega_0^2 X_r^\mu X_{a\mu} + \omega_1^2 \psi_r^A \psi_a^A - \Omega_0 \dot{X}_r^\mu X_{a\mu} - \Omega_1 \dot{\psi}_r^A \psi_a^A + \dots \tag{59}$$

Importantly, to mimic what happens in the holographic models we need to break explicitly shift symmetry and spacetime translations down to their diagonal subgroup. This could be only achieved by asking that

$$\omega_0 = \omega_1, \qquad \Omega_0 = \Omega_1, \tag{60}$$

and the structure of the symmetry breaking term is more constrained and given by:

$$\mathcal{L}_{breaking} = \omega_0^2 \left( X_r^\mu X_{a\mu} + \psi_r^A \psi_a^A \right) - \Omega_0 (\dot{X}_r^\mu X_{a\mu} + \dot{\psi}_r^A \psi_a^A) \dots \tag{61}$$

Given the structure of this term, this will introduce a constant "mass" term $\omega_0^2$ in both the dispersion relation of the phonons and of the phason. The phonon dispersion relation is now

$$\omega^2 + i\,\Omega_0\,\omega = \omega_0^2 + \mathcal{V}^2\,k^2\,, \tag{62}$$

where the symbol $\mathcal{V}$ determines the generic speed of propagation (which will be different accordingly to the direction of propagation). The original massless Goldstone modes acquire a finite mass $\omega_0^2$ and damping $\Omega_0$ as expected. In the same spirit, the dispersion relation for the phason becomes

$$\omega^2 + i\,\bar{\gamma}\,\omega = v^2 k^2 + \omega_0^2\,, \tag{63}$$

where $\bar{\gamma} \equiv \gamma + \Omega_0$ and $\gamma$ is the ordinary phason damping coefficient. Notice that the same "mass" term appears for the phonon and the phason. Expanding the solution at small momenta and small explicit breaking, $\omega_0 \ll 1$, the diffusive phason acquires a finite damping $\Omega$:

$$\omega \equiv -i\,\Omega + \mathcal{O}(k^2) = -i\,\frac{\omega_0^2}{\bar{\gamma}} + \mathcal{O}(k^2)\,, \tag{64}$$

which is determined by

$$\Omega \equiv \frac{\omega_0^2}{\bar{\gamma}}\,. \tag{65}$$

Using the fact that the diffusive constant of the phason is given by $D = v^2/\bar{\gamma}$, we finally find that

$$\Omega = \frac{\omega_0^2 D}{v^2} + \dots\,, \tag{66}$$

which can be compared directly with the holographic results and in particular with the conjecture of [41]. The $\dots$ include all the corrections which are higher order in the explicit breaking scale. Borrowing the notations of [41], we can write

$$\Omega = G\,m^2\,\Xi = \chi_{\pi\pi}\,\omega_0^2\,\Xi\,, \tag{67}$$

which, using $D = G\,\Xi$, becomes

$$\Omega = \chi_{\pi\pi}\,\omega_0^2\,\frac{D}{G} = \frac{\omega_0^2 D}{v^2} = \frac{\omega_0^2 D}{\mathcal{V}^2}\,, \tag{68}$$

as derived above.

In this last step, we have used a very non-trivial relation between phason elasticity and phonons elasticity which arises (again) because of the intertwined symmetry breaking pattern. As already stressed several times, due to the preservation of the diagonal subgroup between spacetime translations and internal phason shifts, a combination of the two transformations leaves the system invariant. In other words, the diagonal group $D \equiv (x \to x + a) \times (\phi \to \phi - a)$ is never broken, nor spontaneously nor explicitly. Now, notice that a transformation of the type $(\phi \to \phi + b)$ is a phason shift with its associated phason elastic modulus, while $(x \to x + a)$ is a standard phonon displacement associated to the common elastic moduli. In the end, this implies a close relation between the phasons and the phonons elastic moduli which leads to the identification of the phonons and phasons speed, $v = \mathcal{V}$. Not surprisingly, this interplay can be verified directly in the holographic models with pure SSB of translations and internal shifts [107, 108]. From the holographic perspective, a phonon shift is associated to spacetime translations and it is encoded in the shear component of the graviton, $h_{xy}$. The response in the stress tensor $T_{xy}$ proportional to that is the phonon shear modulus. Nevertheless, one can also engineer a phason shift by simply using a scalars configuration of the type $\phi^I = \alpha\,\epsilon_{IJ}\,x^J$. This time, the response in the stress tensor characterizes the phasons elastic modulus. Performing

these procedure, one finds that the response to a spacetime translations is exactly the opposite of that to a phason shift, which is nothing else than the statement that the diagonal group is preserved. Importantly, this feature is already encoded in the Stueckelberg nature of the scalars $\phi^I$ and can be understood in terms of gravitational dynamics by looking at the gauge invariant "strain", which is indeed a combination of metric strain and scalar strain.

Finally, it is interesting to notice that this universal relation has already appeared implicitly in the context of incommensurate structures in [81].

In summary, the universal relation found in the holographic models can be robustly derived using EFT techniques at finite temperature. Moreover, the fact that the relaxation rate of the phason is proportional to the pinning mass of the phonons is clearly related to the symmetry breaking pattern preserving the diagonal of phase shifts and spacetime translations. In case the two breakings were completely independent and not induced by the same breaking term, those two quantities would not be related by any means.

## 3   Discussion

In this work we have built a finite temperature effective field theory for quasicrystals starting from an action principle and exploiting solely the symmetries of the system. Our motivation was to provide a deeper and formal understanding of the diffusive nature of the phason mode at large wavelengths, which goes beyond the heuristic and phenomenological arguments spread in the literature. Employing Schwinger-Keldysh techniques and the superspace formalism, we derived the diffusive-to-propagating dispersion relation of the phason mode typical of quasicrystals and observed experimentally. From a technical perspective, the diffusive dynamics is a direct consequence of the fact that phason shifts are symmetries of the system with no associated Noether current nor conserved charge. This is consistent with the EFT requirements only at finite temperature and in the absence of periodicity. The curious fact that phason shifts are not standardly realized symmetries coincides with the statement that they leave the free energy invariant but they do not commute with the hamiltonian of the system.

To the best of our knowledge, our work represents the first description of quasicrystals from an action principle and it is complementary to the older hydrodynamic formulation of [43]. Moreover, it provides a robust field theory explanation to the recent holographic results [34] revealing the existence of a diffusive Goldstone mode associated to the spontaneous breaking of a global bulk symmetry.

Finally, using the Schwinger-Keldysh techniques for states with broken translations, we were able to derive formally the universal relation between the phason relaxation rate and the pinning mass of the phonons found in holography [41]. From our construction, it appears clear that this relation is a direct manifestation of the locked-in breaking of spacetime translations and phase shifts. How general and necessary is to retain this symmetry breaking pattern, with their diagonal subgroup preserved, it is not clear to us and it definitely represents an important open question.

Our results represent another beautiful application of the recent finite temperature EFT formalism, which extends the previous techniques to the more natural and ubiquitous finite-temperature systems. Despite the practical use for quasicrystal systems, there are several formal questions floating around our discussion such as the role of the Goldstone theorem for

dissipative finite-temperature systems, the nature of the corresponding Goldstone bosons and the boundary interpretation of global symmetries in the bulk within the holographic constructions.

As always, there are several questions left for the future. Among them, it would be interesting to study the consequences of gauging the phason shifts in relation also to the holographic model of [102], to understand the propagative nature of the phason dynamics in modulated structures and to formalize from an action principle the physics of incommensurate phases and the commensurate-incommensurate transition at finite temperature.

We conclude by informing Charles de Gaulle that the only way to govern a country which has 246 varieties of cheese is to use effective field theories.

# Acknowledgements

We thank D.Chester, F.Fang, K.Irwin, A.Esposito, S.Grieninger, A.Nicolis, B.Gouteraux, A.Donos and A.Zaccone for enjoyable discussions and useful suggestions. M.B. acknowledges the support of the Spanish MINECO "Centro de Excelencia Severo Ochoa" Programme under grant SEV-2012-0249. Michael Landry acknowledges support from the US Department of Energy grant DE-SC0011941.

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
