# Peer review of "Effective Field Theory for Quasicrystals and Phasons Dynamics"

_SciPost Physics, doi:SciPost Phys. 9, 062 (2020)_

## Round 1 · Referee Report · Anonymous (Referee 1) · 2020-9-27

Strengths

  1. The authors construct for the first time an effective action for quasicristals using the recently developed Schwinger-Keldysh effective field theory techniques. This method surpasses earlier phenomenological treatments and can be further used to study other universal properties of quasicrystals.

  2. Using the same Schwinger-Keldysh effective field theory techniques in the case of broken translations, the authors were able to derive a relation between the phason relaxation rate and the pinning mass of the phonon recently observed in holographic systems. The derivation provides strong evidence of the universality of this relation.

Weaknesses

  1. A point that has not been addressed in the paper is that of higher-order derivative corrections, a natural implementation of effective field theory techniques. The authors write a Lagrangian to leading order in a derivative expansion and derive (52)-(54) as well as (66). How would these results change in the presence of higher derivative terms in the Lagrangian?

  2. Moreover, are there constraints on the coefficients appearing in (38) coming from the requirement of convergence of the path integral? If so, how are they affecting relations like (54) and (66)?

Report

Given the original material presented in this paper and the clear exposition, I recommend this paper for publication in SciPost with minor changes enumerated below.

Requested changes

  1. In the first paragraph on page 3, reference [12] seem to have appeared earlier than [11] while the text is phrased otherwise.

  2. In the last paragraph on page 3, what are the "diffusive Goldstone bosons"? Is this terminology equivalent to Type II Goldstone bosons mentioned in the previous paragraph?

  3. In equation 26 there appears to be a typo since what appears in the equation does not correspond to what is written in the text that follows.

  4. At the beginning of the paragraph containing (38), $\partial_{\mu}\psi^i$ appears. Is this $\psi^i_r$ or $\psi^i_a$?

  5. Above (38) it is said that $\partial_{\mu}\psi^4_r$ has a vanishing expectation value. Why is that the case?

  6. Below (43), the sentence "The fact that $T^{\mu\nu}... $" seems to be incomplete.

  7. Eq. (55) is understood as the equation of motion associated with the field $\psi^4$. It is also understood that the shift symmetry (31) leads to a vanishing Noether current. However, the link between these two statements is not clear. In particular claims such as"Thus, the fact that at low momentum the phason is diffusive is a direct result of the absent Noether current associated... which confirms explicitly the previous arguments." are not evident.

  • validity: good
  • significance: good
  • originality: good
  • clarity: good
  • formatting: good
  • grammar: good

Author:  Matteo Baggioli  on 2020-10-07  [id 997]

(in reply to Report 1 on 2020-09-27)

The reply to the reports of the two referees can be found in the attachment.

Attachment:

Breaking_Bad_Symmetries.pdf

---

## Round 1 · Referee Report · Anonymous (Referee 2) · 2020-10-5

Strengths

  1. The paper constructs an effective field theory (EFT) for quasicrystals using a recently developed framework based on the Schwinger-Keldysh formalism. This enables the authors to completely characterize the low-energy dynamics of the system in terms of a set of symmetries.

  2. Within this general framework, the paper further derives a relation, recently found in holography, between the relaxation rate of the phason and the pinning mass of the phonons. This provides support that such relation is universal.

Weaknesses

The paper did not comment on how the EFT generalizes to more general setups, such as when dropping Poincare' invariance and the $SO(4)$ quasicrystal symmetry.

Report

The paper is well-written: it includes a nice discussion of the overall context, and most steps of the construction are well explained and articulated. I therefore recommend this paper for publication in SciPost, after the minor requests of change below have been addressed.

Requested changes

  1. The phason dispersion relation is obtained by performing a split of the quasicrystal modes $\psi^A$ ($A=1,\dots,4$) into $\psi^i$ ($i=1,2,3$) and $\psi^4$. In particular, the authors first solve for the dynamics of the quasicrystal fields $\psi^i$ ($i=1,2,3$), which leads to a partial fix the worldvolume diffeomorphism symmetry, and subsequently solve for $\psi^4$. Is a splitting of the $\psi^A$ fields necessary in order to obtain the phason dispersion relation, or does this relation arise also when solving the equations in a manifestly $SO(4)$-invariant way? It may be good to include a comment on this.

  2. I assume that Poincare' invariance is adopted for simplicity. It would be worth to mention that this assumption can be relaxed, since normally these systems do not enjoy Poincare' symmetry.

  3. On a related note, below eq. (44) it is mentioned that stress-energy conservation is a consequence of gauging Poincare' symmetry. It would be more precise to say that stress-energy conservation is a consequence of gauging spacetime translation symmetry, which is a slightly different statement (for example, the associated background would in general not be a spacetime metric, unlike in the Poincare' case).

Finally, I found a few typos: -Above eq. (47): $\psi^A_{r/a}$ -> $\psi^i_{r/a}$ -Above eq. (48): "fluid world'' volume" -> "fluid worldvolume''" -Beginning of sec. 2.4 "quaiscrystal" -> "quasicrystal" -Eq. (59), $X_r^\mu X_r^\mu$ -> $X_r^\mu X_{a\mu}$, and $\psi_r^A \psi_r^A$ -> $\psi_r^A \psi_a^A$

---

## Round 2 · Referee Report · Anonymous · 2020-10-16

Report

My remarks have been fully addressed. I therefore recommend this paper for publication in SciPost.

---

## Round 2 · Referee Report · Anonymous · 2020-10-16

Report

The authors have addressed the questions raised in the previous report. Moreover, given the originality and the clear presentation, I recommend this manuscript for publication in SciPost.

  • validity: -
  • significance: -
  • originality: -
  • clarity: -
  • formatting: -
  • grammar: -

Author:  Matteo Baggioli  on 2020-10-16  [id 1010]

(in reply to Report 1 on 2020-10-16)

We would like to thank the referee for his/her time and for the positive evaluation of our work.

---

## Round 2 · Author Response

The reply and the list of changes are in attachment to the authors reply to the referees.

---

## Editorial Decision

published